# Establishing Diagnostic and Differential Diagnostic Criteria for Amyotrophic Lateral Sclerosis

**DOI:** 10.3390/jcm15010287

**Published:** 2025-12-30

**Authors:** Edyta Dziadkowiak, Karol Marschollek, Anna Kwaśniak-Nowakowska, Anna Zimny, Wiktoria Rałowska-Gmoch, Małgorzata Boroń, Magdalena Koszewicz

**Affiliations:** 1Clinical Department of Neurology, University Centre of Neurology and Neurosurgery, Faculty of Medicine, Wroclaw Medical University, Borowska 213, 50-556 Wroclaw, Poland; edyta.dziadkowiak@umw.edu.pl (E.D.); malgorzataboron4@wp.pl (M.B.); 2Department of Neurology with Stroke Unit, T. Marciniak Lower Silesian Specialist Hospital—Emergency Medicine Center, 2 Gen. Augusta Emil Fieldorf Street, 54-049 Wrocław, Poland; annakwasniak@interia.pl; 3Department of General and Interventional Radiology and Neuroradiology, Wroclaw Medical University, Borowska 213, 50-556 Wroclaw, Poland; anna.zimny@umw.edu.pl; 4Department of Neurology, St. Jadwiga Provincial Specialist Hospital, Institute of Medical Sciences, University of Opole, pl. Kopernika 11a, 45-040 Opole, Poland; wiktoria.ralowska-gmoch@uni.opole.pl; 5Clinical Neurophysiology Laboratory, University Centre of Neurology and Neurosurgery, Faculty of Medicine, Wroclaw Medical University, Borowska 213, 50-556 Wroclaw, Poland; magdalena.koszewicz@umw.edu.pl

**Keywords:** motor neuron disease (MND), amyotrophic lateral sclerosis (ALS), atypical MND-ALS variants, superoxide dismutase 1 (SOD1) mutation

## Abstract

Motor neuron disease (MND) represents a broad and heterogeneous group of disorders involving the upper or lower motor neurons, represented mainly by amyotrophic lateral sclerosis (ALS), primary lateral sclerosis (PLS), progressive muscular atrophy (PMA) and progressive bulbar palsy (PBP). Primary motor neuronopathies are characterized by progressive degenerative loss of anterior horn cell motoneurons (lower motor neurons) or loss of giant pyramidal Betz cells (upper motor neurons). Rare atypical variants of MND-ALS include flail arm syndrome (FA), flail leg syndrome (FL), facial-onset sensory and motor neuronopathy (FOSMN), finger extension weakness and downbeat nystagmus motor neuron disease (FEWDON-MND) and long-standing and juvenile MND-ALS. In this article, we present a review of diagnostic criteria and the differential diagnosis for MND, focusing on ALS.

## 1. Introduction

Motor neuron disease (MND) is clinically heterogeneous. The clinical picture is dominated by paresis, resulting from motor pathway involvement by means of a progressive degenerative process [1,2,3]. The most common motor neuron disease is amyotrophic lateral sclerosis (ALS), in the former literature referred to as Charcot or Lou Gehrig’s disease. Clinically distinct forms of MND include classic ALS, primary lateral sclerosis (PLS), progressive muscular atrophy (PMA), progressive bulbar atrophy (PBP), flail arm syndrome (FA), flail leg syndrome (FL) and the hemiplegic form (Mills syndrome) [4,5]. Atypical variants of MND-ALS also include facial-onset sensory and motor neuronopathy (FOSMN), finger extension weakness and downbeat nystagmus syndrome (FEWDON-MND), and long-standing and juvenile MND/ALS [6]. Currently, primary lateral sclerosis is classified as a separate disease entity. Underlying all of these is the same pathophysiological process—loss of anterior horn motor neurons (lower motor neurons, LMNs) or loss of Betz’s giant pyramidal cells (upper motor neurons, UMNs) [1,7].

ALS is a neurodegenerative disease whose axial clinical manifestation is central and peripheral motor neuron damage. It occurs with a frequency of 2/100,000, most commonly in the fifth and sixth decades of life, with a median survival time of 3–5 years. The prevalence is higher in men [1,7,8].

The pathological process underlying ALS has been conceptualized as a corticofugal axonal spread model [9], with the progressive accumulation and propagation of pathological aggregates of the 43-kDa phosphorylated transactivation response DNA-binding protein (pTDP-43), which serves as the pathological hallmark in the majority of ALS cases. In this model, neuronal degeneration originates in neocortical pyramidal cells, where the formation of inclusion bodies consisting of ubiquitin protein conjugated with pTDP-43 begins. From the cortex, pTDP-43 pathology is hypothesized to propagate in an anterograde fashion. The pathological proteins spread through axons and across synaptic junctions, consistent with a prion-like mechanism of transneuronal spreading. This process leads to the involvement of the brainstem motor nuclei and anterior horn cells of the spinal cord, corresponding clinically to the characteristic progression of motor weakness in the course of the disease. This pattern of propagation seems to be in line with the observations from experimental studies, in which pTDP-43 was shown to propagate along corticofugal pathways, inducing ALS-like pathology in mice [10].

According to Braak et al. [9], the natural spread of neuropathological changes in ALS can be divided into four stages (Figure 1). In Stage 1, lesions are located in the motor cortex, bulbar and spinal somatomotor neurons. In Stage 2 lesions are shown spreading to the reticular formation, precerebellar nuclei and thalamus, in Stage 3 to the postcentral neocortex and striatum, while Stage 4 affects the hippocampal area. This progression supports the hypothesis that ALS pathology spreads in a topographically ordered sequence, which corresponds to the usually observed clinical evolution.

In the early stages of the disease, there is usually paresis of the distal muscles of one limb. Patients then report difficulty with precise hand movements or weakness of the foot with foot drop. Less commonly, the disease begins with swallowing and/or speech disorder. As the disease progresses, paresis and muscle atrophy extend to subsequent limbs and/or the face. With initial weakness and atrophy of the small muscles of the hand, symptoms gradually spread to the unilateral muscles of the arm, then to the unilateral lower limb or bulbar area. When the lower limb is involved, the paresis successively involves the muscles of the contralateral lower limb or the muscles of the upper limbs. When the first symptoms are related to the bulbar symptoms, the muscles of the upper limbs are affected next, followed by the muscles of the lower limbs. Bulbar and pseudobulbar symptoms in the form of dysarthria and dysphagia, as well as respiratory muscle weakness, are observed at different stages of the disease, but most often at an advanced stage.

Neurological examination reveals exaggerated deep tendon reflexes in the initial stage of the disease, and weakened reflexes in the advanced stage, as well as the bilateral Babinski sign, fasciculations of the limbs, trunk and tongue muscles. In the initial stage, sensory disturbances and dysautonomia symptoms are usually absent, and the sensory organs and intellect are spared. Muscle paresis and atrophy may be asymmetrical. The course of the disease is not linear. A period of deterioration may be followed by a stable state [7,8].

Although several reviews addressing the diagnostic approach to MND have been published in the preceding years, this review aims to provide a complementary perspective. Recent reviews predominantly focus on the diagnosis of ALS, including discussions of available diagnostic tools and emerging biomarkers [11,12], as well as diagnostic delays [13]. This paper intends to put a special emphasis on the practical interpretation of electrophysiological criteria and how these criteria help differentiate various forms of MND. Moreover, the authors wish to provide a comprehensive and clinically oriented overview of the differential diagnosis of ALS, which could support a more accurate and efficient diagnostic process in everyday practice.

## 2. Materials and Methods

A literature search, covering the period of 1990–2025, was performed using the PubMed and Embase databases. Due to the narrative character of the review, no strict, formal search strategy or study selection protocol were applied. The selection intended to include representative publications that inform current clinical practice.

## 3. Clinical Spectrum of Amyotrophic Lateral Sclerosis

### 3.1. Progressive Muscular Atrophy (PMA)

Pure involvement of the lower motor neuron with onset in adulthood and a relatively benign course. Muscle atrophy and weakness can be symmetrical or asymmetrical, but always bilateral. The atrophy usually appears earliest in the thenar eminence of the thumb, followed by the other small muscles of the hand. Subsequently, it usually involves the muscles of the shoulder girdle and/or the muscles of the forearms and shoulders symmetrically. Atrophied muscles show bundle tremors, often extensive, including muscles in which atrophy is not yet apparent. As the disease progresses, paresis of the atrophied muscles increases, the earliest being the distal muscles, leading to a ‘claw’ or ‘monkey’ position of the fingers. Initially, periosteal and tendon reflexes are weakened, then abolished. Paresis or paralysis involves the entire upper limbs and then the lower limbs. Four-limb paralysis occurs, with low muscle tone and the absence of deep reflexes. Respiratory failure and then respiratory paralysis occur. The described pattern of PMA occurs when the pathological process starts in the cervical region of the spinal cord and spreads caudally. However, cases are observed where the initial symptoms of the disease occur in the distal muscles of the lower limbs. The clinical syndrome may mimic peroneal nerve damage. In PMA, as defined, there are no symptoms of upper motor neuron damage [14,15].

### 3.2. Progressive Bulbar Atrophy (PBP)

Damage to the lower motor neurons in the brainstem (nuclei of nerves V–XII). The incidence is difficult to assess, estimated at 5–20% of cases. PBP is more common in women. The median survival time is shorter than in other motor neuron diseases, ranging from 3 months to 6 years from the onset of the first symptoms. Although the phenotype is named after damage to the inferior motor neuron, the clinical picture shows signs of pseudobulbar syndrome (damage to the corticobulbar tracts) in addition to symptoms of brainstem damage [16,17]. Symptoms of PBP include disorders of articulation, chewing and swallowing. Characteristically, there is no spinal cord involvement for the first 6 months after the onset of symptoms. On examination, tongue muscle atrophy and fasciculations, tongue deviation, impaired palatal and tongue movements and sometimes peripheral paresis of the VII nerve can be found. Central paresis of nerve VII, a brisk jaw jerk and laughter or forced crying indicate the involvement of the pyramidal tract in the disease process. As the disease progresses, dysarthria and dysphagia develop. This presentation requires differentiation from myasthenia gravis and brainstem stroke [6,17].

### 3.3. Flail Arm Syndrome (FA, Flail Arm, Brachial Amyotrophic Diplegia, Vulpian–Bernhardt Syndrome)

Is a variant of MND that is characterized by predominantly proximal upper limb weakness and atrophy. Due to its heterogeneous presentation and relatively slow progression, differential diagnosis can be difficult, especially in the early stages. The disease predominantly affects younger men (4–10:1 M:W) compared to the classic form of MND and survival time from diagnosis is longer than 5 years. Most patients develop symmetrical and predominantly proximal weakness and atrophy of the upper limbs. In the FA phenotype, according to Braak’s theory and spinal spread of symptoms, there is involvement of the paraspinal and diaphragm muscles and development of respiratory failure [18,19]. FA syndrome should be differentiated from multifocal motor neuropathy with conduction block (MMN-CB), paraneoplastic syndrome, Kennedy’s disease and spinal muscular atrophy (SMA) of late onset [20].

### 3.4. Flail Leg Syndrome (FL)

Is a rare phenotype of ALS characterized by a slowly progressive course, with a longer survival period than classic ALS. The clinical picture is characterized by symptoms of LMN damage confined to the lumbosacral region for an extended period of time, ranging from 12 to 24 months. The so-called triad of symptoms includes distal lower limb paresis with muscle atrophy and hyporeflexia/areflexia. The longer the disease is confined to the lumbosacral region, the longer the survival. The development of respiratory failure is invariably rare, presented in the literature as case reports [6,21,22,23].

### 3.5. Amyotrophic Lateral Sclerosis, Hemiplegic Form (Mills Syndrome)

A rare (1.2 cases per 1 million population), acquired motor neuron disease, first described in 1900, characterized by unilateral, ascending or descending monoplegia associated with unilateral or asymmetric atrophy of pyramidal tracts, without sensory disturbance. Approximately 50 middle-aged or elderly patients with spastic monoplegia of the lower limb who subsequently developed paresis of the ipsilateral upper limb, then of the contralateral lower limb, and finally of the contralateral upper limb (the so-called N-pattern of paresis) have been described in the last two decades. Clinicians should think of this syndrome in the differential diagnosis of spinal cord injury [24,25]. Figure 2 shows a schematic representation of motor neuron disease variants.

## 4. Diagnostic Criteria

Diagnostic criteria for ALS were developed through the World Federation of Neurology’s Study Group for Motor Neuron Diseases in 1994 and published as the El Escorial criteria, followed by modifications and publication of the Airlie House criteria [26,27]. In 2006, a group of neurophysiologists proposed a modification of them, the so-called Awaji algorithm [28]. Since the publication of the Awaji Criteria, there have been significant advances in neurophysiological, imaging and genetic studies. The original criteria did not take into account the presence of cognitive and behavioral changes, which are now known to occur in up to 50 percent of people with ALS, including frontotemporal dementia in 15 percent of patients [29]. Now, through a consensus of specialists from the World Federation of Neurology, the International Federation of Clinical Neurophysiology and the Motor Neuron Disease Association, criteria called the Gold Coast criteria have been developed [30,31]. The diagnosis of ALS can be established if the following criteria are fulfilled:
1.Progressive motor impairment documented by history or repeated clinical assessment, preceded by normal motor function,

AND

2.The presence of upper (A) and lower (B) motor neuron dysfunction in at least ONE body region (C) or lower motor neuron dysfunction in at least TWO body regions,3.excluding other disease processes.

(A) Upper motor neuron dysfunction implies at least one of the following:

-increased tendon reflexes, including from the clinically involved muscle or adjacent muscle group.-presence of pathological reflexes, including Hoffmann’s sign, Babinski sign, crossed adductor reflex, or snout reflex-increased spastic muscle tone-weakness of voluntary movement that cannot be attributed to lower motor neuron damage or Parkinson’s disease

(B) Lower motor neuron dysfunction in a given muscle requires either:

Clinical examination evidence of muscle weakness and muscle atrophy

or

EMG abnormalities that must include both:

-chronic neurogenic lesions, defined by motor unit potentials of prolonged duration and/or increased amplitude, with unit instability considered to confirm the diagnosis, but not necessary-and features of denervation, including fibrillation potentials, positive sharp waves or fasciculations

(C) Body regions are defined as bulbar, cervical, thoracic and lumbosacral. In order to speak of lower neural involvement, two limb muscles innervated by different roots and nerves must be involved, or it must be noted in one bulbar muscle, or in one thoracic muscle based on physical examination or changes in the EMG.

Table 1 shows a comparative overview of the revised El Escorial, Awaji and Gold Coast diagnostic criteria for ALS.

ALS is a progressive motor impairment most often with a focal onset of symptoms, but a generalized onset is also possible. Motor dysfunction in ALS reflects dysfunction of both lower and upper motor neurons, but it is recognized that upper motor neuron symptoms are not always clinically apparent. Evidence of lower motor neuron dysfunction can be obtained from clinical examination and/or electromyography (EMG) testing. Currently, for diagnostic purposes, dysfunction of the upper motor neuron is established by clinical examination [30,31].

In 30% of patients with ALS, brain MRI may reveal a high signal on T2-weighted and FLAIR images within the pyramidal tracts, seen earliest in the posterior limbs of the internal capsules, and later in the course of the disease in the entire spinal tracts from the precentral gyri to the spinal cord [32] (Figure 3a–d and Figure 4a,b,d,e). Another imaging sign is a hypointensity of the primary motor cortex on SWI or T2* images known as the “motor band sign” (Figure 3e and Figure 4c,f). The sign is produced by iron accumulation within microglia in the motor cortex as they phagocytose degenerating neurons [33]. It may be seen in patients without a hyperintensity of the pyramidal tracts, most often bilaterally, but a unilateral pattern has also been reported. In ALS, this sign is seen along the entire primary motor cortex and in PBP the signal change may only be seen laterally [34]. All imaging symptoms may be very subtle in the initial stage of the disease and progress in time (Figure 4).

In patients without structural lesions visible on MRI and DTI, the parameter of fractional anisotropy (FA) has been found to be useful in the differentiation of ALS patients from healthy controls. In ALS subjects, FA was shown to be significantly decreased in the brainstem (cerebral peduncles, pons and pyramids) as well as within both posterior limbs of the internal capsules [35] (Figure 3f). The DTI study including the tract-of-interest-based analysis showed the same microstructural corticoefferent involvement patterns as in ALS in PBP [16], flail arm syndrome and flail leg syndrome [36] and primary lateral sclerosis [37].

Additional evidence of lower motor neuron dysfunction can be obtained from ultrasound detection of multiple muscle fasciculations [38]. Additional evidence of upper motor neuron dysfunction is shown by the results of transcranial magnetic stimulation of the central nervous system and determination of neurofilament levels in the cerebrospinal fluid [39]. It should be emphasized that these tests are not necessary for diagnosis. ALS may include cognitive, behavioral and/or psychiatric abnormalities, although these are not relevant to the diagnostic criteria [30,40].

## 5. Electrophysiologic Criteria for Motor Neuron Disease Types

Electrophysiologic examinations serve an important role in the diagnosis of MND, helping to distinguish between different subtypes based on patterns of nerve and muscle involvement. By utilizing techniques such as nerve conduction studies and electromyography, clinicians can assess motor unit integrity, detect denervation and rule out mimicking conditions. Each phenotype of MND may present with distinct electrophysiologic findings. Understanding these differences is essential for accurate diagnosis and disease classification, therefore guiding prognosis and management strategies.

### 5.1. Amyotrophic Lateral Sclerosis

Electrophysiologic changes in ALS may be detected before the clinically visible symptoms of muscle weakness; therefore, they play a crucial role in establishing lower motor neuron dysfunction. EMG may reveal spontaneous activity of affected muscles very early in the course of the disease. Characteristic findings indicative of ongoing denervation include fasciculation potentials, fibrillation potentials and positive sharp waves [41]. Ongoing denervation is identified on needle EMG by fibrillation potentials, positive sharp waves and fasciculation potentials, all representing abnormal spontaneous activity in resting muscle. Fibrillation potentials—together with their monophasic counterpart, positive sharp waves—arise from spontaneous depolarization of denervated muscle fibers. This results from increased acetylcholine receptor density on the muscle membrane, which makes fibers hypersensitive after loss of neural input. The amplitude of fibrillation potentials peaks around 3–4 months after denervation and may remain elevated for years, indicating persistent membrane instability. Additionally, motor unit potentials (MUPs) exhibit increased amplitude, polyphasia and prolonged duration due to chronic reinnervation, along with a reduced recruitment pattern reflecting motor unit loss. Needle EMG in ALS shows evidence of chronic denervation, including increased MUP duration due to collateral sprouting, and increased polyphasicity reflecting dyssynchronous reinnervation. Amplitude is often increased as motor units expand, though very small MUPs may appear when axons fail. Decreased recruitment, with fewer MUPs firing at high contraction, is an early indicator of lower motor neuron involvement. MUP instability, seen as variability in peaks or amplitude between potentials, may indicate rapid motor unit loss but is not specific to ALS. Importantly, these abnormalities are widespread, affecting multiple limb and bulbar muscles, consistent with the multisegmental motor neuron involvement characteristic of ALS [31].

Motor NCSs typically demonstrate reduced compound muscle action potential (CMAP) amplitudes caused by motor axon loss. However, these changes are not typically visible in the early course of the disease due to sufficient reinnervation; therefore, their diagnostic utility is limited. One of the most promising ALS electrophysiological biomarkers may be the “split-hand phenomenon”, which is defined as the preferential wasting of the thenar muscles, specifically the abductor pollicis brevis (APB) and first dorsal interosseous (FDI), and sparing of the hypothenar muscle, the abductor digiti minimi (ADM). In NCSs, it is assessed by dividing the compound muscle amplitude potential (CMAP) amplitudes of the aforementioned hand muscles; an APB/ADM ratio < 0.6 or a FDI/ADM ratio < 0.9 are considered pathological [42]. In the meta-analysis of Hu et al. [43], including five studies and 339 patients, split-hand phenomenon was present in 50% of the ALS participants. A split-hand index (SI), calculated by = (APBCMAP × FDICMAP)/ADMCMAP, demonstrated a sensitivity and specificity of 78% and 81%, respectively, for differentiating ALS from neuromuscular and healthy controls. The high diagnostic utility of split-hand phenomenon in ALS may be explained by the primary involvement of the motor cortex and upper motor neuron dysfunction, as the preferential atrophy in muscles innervated by the same myotomes strongly suggests a cortical pathomechanism [44].

Other tools useful both in the diagnostic process and in monitoring disease progression include the Neurophysiologic Index (NI), a simple and reproducible measure that reflects denervation and reinnervation dynamics, which can be calculated from the CMAP, the distal motor latency (DML) and the F-wave frequency ((CMAP amplitude/DML) × F frequency %) [45], and the Motor Unit Number Estimate (MUNE), which quantifies the number of functional motor units by dividing the maximum CMAP amplitude by the average single motor unit potential (MUP), while the Motor Unit Number Index (MUNIX) uses surface EMG data and CMAP amplitude to estimate relative motor unit changes [46]. Such indices proved to be useful in detecting presymptomatic motor neuron loss in ALS patients [47,48], as well as in predicting survival [49].

Artificial intelligence (AI) may support the diagnostic process of MND, particularly through the capability of analyzing multimodal data, combining the value of clinical, laboratory, electrophysiological and other biomarkers. Although AI-based methods are only beginning to be implemented and need validation, preliminary results indicate that AI-driven analysis has the potential to become a significant complement to traditional diagnostic tools.

In one study, machine learning models analyzing F-wave responses were able to predict ALS with meaningful precision and accuracy. Moreover, classification probabilities for ALS patients were statistically different from the diagnoses mimicking ALS symptoms [50]. Chia et al. [51] identified 33 proteins, the levels of which were different in patients with amyotrophic lateral sclerosis (ALS) compared to controls, and applied machine-learning models to create a plasma protein signature that distinguished ALS cases with high diagnostic accuracy, even before the clinical onset of symptoms.

Finally, a systematic review and meta-analysis conducted by Umar et al. [52] assessed the existing studies using AI for the diagnosis of ALS. AI models have demonstrated high performance in the early detection and classification of ALS, with pooled sensitivity and specificity exceeding 90%, particularly when analyzing gait signals and electromyography parameters, with slightly worse performance regarding MR imaging.

### 5.2. Progressive Muscular Atrophy

In PMA, LMN damage occurs in the absence of clinical signs of UMN damage. Electrophysiological studies do not show significant differences compared with ALS [53]. In fact, distinguishing between these two types of MND may be clinically and diagnostically challenging, and autopsy studies often show degenerative changes in the central nervous system despite the clinical diagnosis of PMA [54]. Neurogenic changes are almost always found bilaterally, although in very rare cases they may affect only one side of the body [55].

### 5.3. Progressive Bulbar Palsy

PBP refers to patients who have isolated bulbar involvement within the first 6 months of the disease, and therefore in the diagnostic process significant changes can be detected in the muscles of the oral cavity, pharynx, larynx and tongue, as well as the sternocleidomastoid and trapezius muscles. Repeated electrophysiological studies may be helpful in monitoring degenerative changes in the limbs, as well as in distinguishing PBP from isolated bulbar palsy (IBP), in which limb involvement occurs later and a milder course of the disease is observed. Zhang et al. [56] compared patients with PBP and IBP, establishing a cut-off value for time-to-limb involvement at 20 months. Despite the significantly longer survival time in patients with IBP, no differences were found in terms of neurogenic abnormalities in the tongue, sternocleidomastoid or upper trapezius muscles on EMG. However, in another study on the Chinese population, where IBP was defined according to the 6-month criterion, a significantly more frequent occurrence of neurogenic changes in the abovementioned muscles was found [57]. The presence of subclinical electrophysiological changes does not appear to be significant in the context of disease progression—in one study almost all PBP patients progressed to ALS regardless of the presence of generalized denervation on EMG of the limbs [58].

### 5.4. Flail Arm Syndrome

FA is a variant of the disease that affects only the upper limbs. Muscle examination typically shows symmetrical, bilateral involvement, mainly of the proximal muscles, consistent with LMN damage. However, patients with FA can initially show asymmetric and extensive distal distribution of symptoms. Moreover, in more than half of them, pathologic spontaneous activity (PSA) in both the upper and lower limbs can be found [59]. Some authors underline the potential differences in electrophysiologic patterns between FA and ALS. In a study by Yang et al. [60], patients with FA had significantly lower split-hand index (SI) and resting motor threshold (RMT) compared to the ALS group. However, in another publication comparing FA with upper-limb-onset ALS, despite differences in muscle weakness distribution and a low rate of fasciculation, no remarkable differences were spotted in electromyography [19]. A study on four patients with FA showed not only changes in needle electromyography (denervation in large and widened motor units, diminished recruitment of motor units), but also some abnormalities regarding sensory and motor nerve conduction in median nerves [61].

### 5.5. Flail Leg Syndrome

FLS is a variant of MND with an onset in the lower limbs. This type is characterized by distal LMN damage, initially mainly from the lumbosacral region. In the later course of the disease, electrophysiological abnormalities also appear at other levels of the spinal cord [21,62]. The nerve conduction studies in FL show reduced CMAP amplitudes without evidence of demyelination or conduction block [63].

### 5.6. Amyotrophic Lateral Sclerosis, Hemiplegic Form (Mills Syndrome)

The hallmark of Mills syndrome is progressive hemiparesis, initially affecting one limb. Due to the limited number of reported cases, data on electrophysiologic findings is limited. Of the three patients described by Zhang et al. [64], two had no abnormalities in muscle examination, whereas in one case only minor chronic denervation in the left dorsal interosseous, left sternocleidomastoid muscle and thoracic paraspinal muscles was detected. In another report of three cases [24], one of the patients developed ALS after 16 years of observation, one patient had no abnormalities in electrophysiologic examination and one patient presented only with mild chronic neurogenic changes without evidence of progression.

### 5.7. Primary Lateral Sclerosis

The diagnosis of PLS is based on the presence of clinical features of upper motor neuron damage in the absence of significant features of lower motor neuron damage at least 4 years after the onset of symptoms (for a certain diagnosis) [65]. Therefore, the use of electrophysiological methods is primarily useful in differential diagnosis, as well as to exclude subclinical lower motor neuron damage, which may suggest ALS. However, the diagnostic consensus assumes the admissibility of the presence of minor denervation in a rare extremity muscle, such as increased insertional activity, fibrillations or positive sharp waves [66]. This is supported by the fact that in previous studies no clinically significant differences were observed in patients diagnosed with PLS in whom these changes were found [67,68]. In some observations, such abnormalities may be present in most PLS patients, but are usually stable in time [69,70]. In another study, a small number of participants with minor changes in EMG met the El Escorial criteria for ALS after several years of follow-up, while in the remaining patients the course of the disease was more severe compared to those without any changes in EMG [71].

## 6. Differential Diagnostics

The differentiation of MND should include many rare conditions which are difficult to diagnose. The first diagnostic step should be to determine the progression of the disease, the distribution of the paresis and the associated symptoms. This will allow an initial diagnosis to be established and additional tests to be planned.

### 6.1. Multifocal Motor Neuropathy

Multifocal motor neuropathy (MMN) is a purely motor neuropathy that occurs about 10 times less frequently than ALS. The onset of the disease generally occurs before the age of 50 (average age of onset is 40 years). Men are affected about three times more often than women [72,73].

The development of the disease is related to the presence of IgM antibodies directed against GM1 gangliosides, which are localized within the motor fibers in the region of Ranvier’s constrictions and paranodally. The autoimmune reaction causes demyelinating peripheral nerve damage. The presence of high-titer antibodies is found in approximately 50–60% of patients with MMN. These antibodies are not specific to this neuropathy and can also be detected in 5–10% of ALS patients, but usually at low titers [74,75].

The disease progresses slowly or in leaps and bounds. In MMN, there is asymmetric and multifocal peripheral nerve damage. In two-thirds of cases, symptoms develop in the upper limbs, mainly distally. In one-third of cases, the disease begins with foot drop. In only 5% of patients, the first symptom of the disease is proximal upper limb paresis [72]. Unlike SLA, in which an entire segment of the spinal cord is affected (muscle paresis results from damage to the myotome), MMN is characterized by damage to single peripheral nerves. Other features that distinguish MMN from ALS are the distribution of paresis in the upper limbs and muscle atrophy. A typical symptom in MMN is a dropping of the fingers and/or hand resulting from dysfunction of the forearm extensors. In contrast, in ALS, the first muscles involved are usually the thenar eminence and intercostal muscles. In MMN, it is notable that there is relatively little atrophy of the muscles affected by paresis, whereas in SLA there is marked muscle atrophy from the onset of the disease. 50% of MMN patients have severe muscle spasms and fasciculations. Deep reflexes are generally impaired. Only in 20% of cases can they be vivid, and in 8% of patients they may even be exaggerated. Sensory fibers are not involved in the course of the disease, but patients may report a slight disturbance of vibratory sensibility in the lower limbs. In MMN, there are no bulbar symptoms [76,77].

MRI of the brachial plexuses is useful in the diagnosis of the disease. In 40–50% of cases, T2-weighted plexus hyperintensity, bundle thickening and plexus postcontrast enhancement are found. The changes described are asymmetrical. In the cerebrospinal fluid, protein levels are moderately increased (<1 g/L) in 30% of MMN patients [78,79]. Patients do not benefit from steroid therapy or therapeutic plasmapheresis. The use of intravenous immunoglobulin is the only documented form of therapy [73,77,78].

The routine protocols of electroneurography recommended for MND are motor median, ulnar, peroneal and tibial study mostly on the ipsilateral side with one motor and sensory nerve on the contralateral side to assess symmetry. Sensory median, ulnar, radial and sural nerves on the clinical side should routinely be examined. Contralateral side and proximal studies should be considered in patients with predominantly lower motor syndromes. Erb’s point is very demanding as supramaximal stimulation can be difficult to achieve and there might be costimulated nerves, or submaximal stimulation is mistaken as conduction block at the proximal side. When looking for conduction block, it is often worthwhile to study additional nerves.

Sensory nerve conduction studies should be normal in ALS and MMN, but it is also known that some other processes may reveal superimposed polyneuropathy. The only exception to normal sensory conduction studies in a motor neuron disorder is X-linked bulbospinal muscular atrophy, which is due to the involvement of the dorsal root ganglion.

The crucial aspect of motor nerve conduction study is to be very careful for any signs of demyelination, prolonged distal latencies and late responses, especially conduction block along motor nerves, particularly at non-entrapment sides. Electrophysiologic evidence of demyelination does not occur in ALS. If it is present, another diagnosis should be considered. Conduction block is typically present distally in MMN. Proximally, a drop in amplitude of more than 50% is never seen in ALS.

In routine examination, three limbs should be included, covering distal and proximal muscles with different nerve and root innervation. Thoracic paraspinal muscles should be included in at least three segments (avoiding Th11-Th12). At least one of the bulbar muscles (masseter, tongue, sternocleidomastoid) should also be included, and more when the bulbar weakness is predominant. The examination of the thoracic paraspinal and craniobulbar muscles differentiates motor neuron disease from cervical or lumbar polyradiculopathy. MND is a myotomal disease, in which individual nerves in the same myotome are not spared, as is observed in MMN [14].

In the differential diagnosis of MMN, the evaluation of treatment response to immuno-modulatory and immunosuppressive agents is essential. Intravenous immunoglobulin (IVIG) is the first-line treatment, with efficacy dependent on dose and frequency; some patients may experience a waning response over time. Subcutaneous immunoglobulin provides similar efficacy with improved tolerability. Other agents, including cyclophosphamide, rituximab, cyclosporine, methotrexate, azathioprine and interferon beta-1a, have shown variable outcomes. Among these, cyclophosphamide and rituximab have shown some clinical benefit in case reports, although supporting evidence is limited. Randomized controlled trials evaluating agents such as mycophenolate mofetil have failed to demonstrate efficacy. Corticosteroids and plasma exchange, effective in CIDP, are not beneficial in MMN [80].

### 6.2. Monomelic Amyotrophy

Monomelic amyotrophy (MA, Hirayama disease) is a very rare disease of the lower motor neuron. It is typically associated with young males from East Asian populations (especially in Japan and India), but reported cases show that it can also occur in individuals from other geographic and ethnic backgrounds. Antonioni et al. describe a case of an Albanian woman with adult-onset symptoms and clinical as well as electrophysiological features consistent with Hirayama disease, despite the absence of characteristic imaging findings. This case shows that Hirayama disease can also occur outside Eastern populations, including in Caucasian patients and with later than typical onset, highlighting the importance of considering it in the differential diagnosis regardless of geographic origin [81].

Its prevalence is unknown. MA is far more common in men (M:W 10:1). Onset of the disease occurs around the age of 20. In the course of MA, there is focal muscular atrophy and paresis of one upper limb. Some patients may develop less severe paresis of the opposite limb. The muscles of the forearm and hand, which are innervated by the C7-Th1 roots, are characteristically involved. Sparing of the brachioradialis muscle (C5-C6 root) is typical. The disease progresses slowly, with symptoms increasing over a period of one to five years before stabilizing. The paresis may be intensified by cold. In some patients, a slight tremor of the limb is observed. There are no sensory, bulbar or pyramidal symptoms in MA [82,83,84,85,86].

Very rarely, monomelic amyotrophy can develop in the lower limb. Paresis and atrophy predominantly affect the gastrocnemius muscle and the hamstring muscle. In these cases, the course of the disease is very mild. The etiology of the disease remains unknown. Vascular damage to the spinal cord, resulting from compression of the arterial vessels by the dura mater during neck flexion, is considered to be the cause. Magnetic resonance imaging of the spine shows spinal cord atrophy and a hyperintense spinal cord signal at the C5-C7 level. On functional examination, dilation of the dura mater venous plexuses on the dorsal surface of the cord is observed during neck flexion. On EMG, neurogenic recording is found from muscles innervated by the C7-Th1 roots (or S1–S2 in the case of lower limb involvement) [83,86,87,88]. Electroneurography tests may be normal or reveal asymmetrically low median or ulnar CMAP amplitudes in the affected hand with slightly slowed median or ulnar conduction velocities depending on the degree of axonal loss. The SNAPs are always preserved. In monomelic amyotrophy the ADM is much weaker than APB, where in typical sporadic ALS split-hand syndrome is observed where the FDI and APB are more affected than ADM. A routine electroneurography motor ulnar and median CMAP amplitudes ADM/APB ratio < 0.6 strongly suggest the diagnosis of monomeric amyotrophy rather than ALS. This ratio may be helpful in cases where it is difficult to distinguish monomelic MA and ALS.

The fibrillation potentials are not common. MUAP potentials are large with prolonged duration and recruitment is reduced [14].

On MRI of the cervical spine, in a neutral position, an abnormal T2-weighted signal of the spinal cord at the site of maximum forward shift without an obvious cause may be seen, while in a flexion position, the posterior dural sac crescent may appear as high signal intensity on T1- and T2-weighted sequences that enhances uniformly on postcontrast T1-weighted images with or without epidural flow voids [89].

### 6.3. O’Sullivan–McLeod Syndrome

O’Sullivan–McLeod syndrome is a rare lower motor neuron disorder characterized by slowly progressive weakness and amyotrophy of the distal parts of upper limbs, primarily affecting the hands. Onset typically occurs in early adulthood, although later-onset cases have also been reported. The syndrome presents without sensory or pyramidal signs. Most patients are males, with a long disease duration and a clinical course that is either stable or slowly progressive over extended periods (up to 20 years) [6,90,91].

Electrophysiological studies usually demonstrate chronic denervation in the C7–T1 myotomes, while nerve conduction studies remain normal. Neuroimaging findings are generally unremarkable, though some patients may show longitudinal T2 hyperintensity in the anterior horn of the cervical spinal cord [91].

To date, no effective treatment or response to immunotherapy has been identified. Despite similarities to Hirayama disease, it is considered a distinct clinical entity [6,90,91].

### 6.4. Acute Motor Axonal Neuropathy

Acute motor axonal neuropathy (AMAN) is a variant of Guillain–Barré syndrome (GBS). The incidence of the axonal form of GBS is low in Europe and North America, but it is the most common variant of the disease found in China. Features include abrupt onset, flaccid ascending symmetrical limb paresis, more severe distally, and the weakness or absence of deep reflexes. In most patients, maximum symptom severity occurs within four weeks. In 85% of cases, the disease is preceded by a gastrointestinal infection caused by Campylobacter jejuni. By means of a mechanism of molecular mimicry between liposaccharides of the bacterial cell wall and molecules on the axon surface, IgG antibodies are produced that are directed against the gangliosides GM1 and Gd1a [92,93].

In earlier nerve conduction studies, AMAN was characterized as axonal motor nerve damage, i.e., a reduction in the amplitude of motor responses. The current literature describes motor conduction blocks without signs of demyelination, an electrophysiological phenomenon defined as temporal CMAP dispersion (motor nerve conduction blocks [CBs]) caused by dysfunction of Ranvier’s nodes (so-called nodopathies). Axonal conduction blocks may be reversible and clinical improvement of the patient is then observed. CBs can also progress to axonal degeneration, associated with a poorer prognosis. Conduction parameters in sensory fibers are normal. The electromyographic parameters indicate neurogenic damage in the muscles examined with numerous denervation potentials [94]. AMAN responds better to treatment with immunoglobulins than plasmapheresis [93,95].

### 6.5. Motor Variant of Chronic Inflammatory Demyelinating Polyneuropathy

The motor variant of chronic inflammatory demyelinating polyneuropathy (CIDP) is very rare and accounts for 7–10% of CIDP cases. The disease develops more frequently in young people (usually before 20 years of age). It is characterized by symmetrical distal and proximal limb paresis, gradually increasing over at least 8 weeks. Sensory symptoms are absent or very discrete [96,97,98].

The cause of the development of CIDP is not sufficiently understood. Both cellular and humoral mechanisms are suspected to be involved. No antibodies are generally detected in the blood serum. Nerve conduction studies reveal damage to motor fibers of the peripheral nerves of a primary demyelinating nature. Conduction parameters in sensory fibers are normal [96,98,99]. The first-line drugs for the treatment of CIDP are immunoglobulins and corticosteroids [96,100].

On an MRI, the hallmarks of CIDP are thickening, T2-hyperintensity and enhancement of peripheral nerves, the brachial and lumbosacral plexus and nerve roots. Approximately one-third of patients have cranial nerve involvement [101].

### 6.6. Genetically Determined Motor Neuron Diseases

#### 6.6.1. Spinal Muscular Atrophy

Spinal muscular atrophy (SMA) is the most common genetically determined motor neuron disease inherited in an autosomal recessive manner, with a progressive neuromuscular disease associated with typically proximal muscle weakness and atrophy due to degeneration of the anterior horn cells of the spinal cord. The preferred method for diagnosing SMA is molecular genomic analysis. SMA is divided into two groups: 5q-SMA and non 5q-SMA. SMA is further divided into five subtypes (0, 1, 2, 3, 4), distinguished by the age of onset of the disease symptoms and the achievement of motor development milestones. This is a simple and practical classification, and therapeutic drugs have only been developed for 5q-SMA (nusinersen, onasemnogene abeparvovec, risdiplam) and not for non-5q-SMA disease [102].

Most patients with SMA have mutations in the survival motor neuron SMN1 gene on chromosome 5q, and the diagnosis is reached when the copy number is zero. SMA type 0 is the most severe type which develops in the fetal period; the infants exhibit respiratory difficulties from birth and require ventilation. SMA type 1 is also characterized by the very severe onset of clinical signs before the age of 6 months and requires ventilation. The SMA 2 gene is homologous to SMN1 and lies within the same chromosome region 5q13 [103].

In SMA type 2, the age of onset is before 18 months; patients can sit but they are unable to stand or walk alone. SMA type 3 divides into type 3a where time of onset ranges from 18 months to 36 months of age, and type 3b where the disease starts at the age of over 36 months. Patients can usually walk, but there is a risk of losing this ability [103,104].

SMA3 and SMA4 progress slowly. Adolescence is very difficult for SMA 3b-type patients, as it is then when they usually lose the ability to walk. Muscle weakness is usually symmetrical, more proximal than distal, with the legs being more affected than the arms. However, the diaphragm, extraocular muscles and facial muscles are relatively preserved [103].

Disease onset in adulthood occurs in type 4 SMA, and it accounts for <5% of SMA cases. Males and females are affected with equal frequency. Symptoms of the disease usually develop after the age of 30 and onset is difficult to detect. Patients report decreased mobility and increased falls. They have difficulty climbing stairs or standing up from a squatting position. The clinical picture is dominated by proximal symmetric paresis, mainly of the lower limbs, with muscle atrophy, spasms and fasciculations [105].

Before genetic testing, EMG was used widely to diagnose SMA. It can be used to confirm the diagnosis and to assess the progression of the disease and the number of destroyed motor neurons. In SMA, fibrillations on the tongue may be present. The electromyographic recordings show neurogenic damage to the examined muscles. Exercise records are depleted, high-voltage, often with amplitudes of >10 mV. Parameter values of motor unit potentials are significantly increased. At rest, fasciculations and denervation potentials (fibrillations and sharp waves) are recorded in the examined muscles. The entire electrophysiological study indicates an ongoing simultaneous process of de- and reinnervation. Motor and sensory conduction in SMA are normal. The diagnosis is based on the result of a targeted genetic test for SMA [106].

Electroneuromiography findings depend on the clinical types of SMA. In infancy (in a severe type—SMA0/SMA1). denervation potentials (fibrillations) are recorded and reinnervation is not observed. Electroneurography should typically be normal, but in SMA1 motor conduction study might be lower because of late myelination, and in rare cases even sensory nerves might be involved. Nerve conduction velocities may be normal until the late stage of the disease, although exceptional cases have been reported with associated sensory neuropathy or sensory ganglionopathy. SMA does not typically involve sensory nerves; thus, sensory nerve conduction studies are expected to be normal, which is similar to ALS.

Denervation is observed in distal and proximal muscles. In more severe types (SMA1 and SMA2), fibrillations and positive waves occur more often than in type 3. The EMG will demonstrate poor recruitment of motor units with reduced MUPs. Moreover, losing motor units is often mistaken as myopathy. Pseudomyotonic discharges often appear in severe types of SMA. In patients with SMA who have had long disease duration, the EMG may also reveal compensatory changes such as reinnervation and an enlargement of MUAP amplitudes.

In SMA 3, the progression is slower compared to other motor neuron disorders. Fibrillation and positive waves are rather rare, while fasciculations are more common and the MUAPs are large (much higher than in ALS), long and polyphasic potentials including satellitars with decreased recruitment which signifies the loss of motor units. The findings are localized mostly in proximal muscles of the legs [107]. Patients with later-onset forms of SMA (types 3 and 4) may have slightly reduced or normal CMAP amplitudes and may not exhibit fibrillation potentials.

In SMA, predominant MRI features are those of muscle atrophy of different severities depending on the type of SMA [108]. In the spinal cord, MRI may reveal atrophy and T2 hyperintensities in the anterior horns of the cervical cord, and to a lesser degree in the thoracolumbar cord, which are thought to correspond histopathologically to motor neuron loss in these regions [34].

With the introduction of nusinersen, risdiplam and onasemnogene abeparvovec, early diagnosis of the disease offers a chance to avoid disability [109].

The existence of non-5q-SMA had already been recognized. It has been very difficult to properly classify non-5q-SMA. Similar symptoms may not be indicative of a single specific disease, and many different disorders with similar symptoms may exist. To achieve an accurate diagnosis, DNA analysis is essential [102,110].

#### 6.6.2. Spinal and Bulbar Muscular Atrophy

Spinal and bulbar muscular atrophy (SBMA; Kennedy disease) is inherited in a recessive manner, coupled to the X chromosome. Patients have an increase in the number of CAG trinucleotide repeats in the first exon of the gene for the androgen receptor [111].

Clinical symptoms result from damage to the motor cells of the anterior horns of the spinal cord and the motor nuclei of the bulb. The characteristic symptom of SBMA is proximal paresis, initially of the lower extremities, accompanied by pain and muscle spasms. At the beginning of the disease, the paresis may have an asymmetric distribution (about 50% of patients). The disease rarely begins with muscle involvement of the bulb region. As the disease progresses, dysarthria, dysphagia and dysphonia develop. Characteristic of the disease are fasciculations in the perioral region and within the tongue, as well as atrophy of the tongue with increased wrinkling and a deep central furrow [112,113]. Symptoms also include upper limb tremor, which precedes the onset of lower limb paresis by up to a decade. Subclinical sensory involvement is found in about 70–100% of patients. On electrophysiological examination, motor nerve conduction parameters are within the reference range, although there may be features of demyelinating peripheral nerve damage. A typical feature is reduced amplitude of sensory responses [4,111].

In addition to motor symptoms, endocrine disorders are also present in SBMA. These include reduced fertility, gynecomastia, impaired glucose metabolism and dyslipidemia [111,114].

Araki et al. analyzed 144 Japanese patients with SBMA to assess myocardial involvement. An abnormal ECG was recorded in 48.6%, with the most common finding being ST-segment elevation in the V1-V3 leads. 17 out of 144 patients (11.8%) were diagnosed with Brugada syndrome [115], a genetic disease based on a mutation of the sodium channel gene. Its main clinical manifestation is sudden cardiac arrest, usually occurring in young people without significant heart disease. Based on an analysis of data from various publications, it has been determined that Brugada syndrome is responsible for 4–12% of cases of unexpected sudden death and for up to 50% of cases of sudden death in people with healthy hearts. The incidence of Brugada syndrome is estimated at 5–66 cases per 10,000 people. Meanwhile, the average age of patients who develop the disease is 35–40, with a range of 6 months to 77 years [116]. The disease affects men far more often than women (M:W-8:1). ECG abnormalities are the hallmark of Brugada syndrome. These include abnormalities of repolarization and depolarization, without specific organic heart disease or other conditions or factors known to lead to ST-segment elevation in right precordial leads. No mutations in the SCNA5, CACNA1C or CACNB2 genes associated with Brugada syndrome were detected in the Japanese patients described above. Myocardial biopsy in seven SBMA patients revealed the accumulation of pathological androgen receptor protein in cell nuclei, confirming myocardial involvement [115].

ALS is a progressive neuromuscular disorder with significant mortality. Patients with this disease die mostly as a result of respiratory failure; cardiovascular causes are increasingly responsible for mortality. A case of type 2 Brugada syndrome causing ventricular tachyarrhythmia and cardiac arrest in a patient with ALS has already been reported [117]; therefore, the cardiovascular assessment needs to be included in ALS.

In contrast to ALS, patients with SBMA show no detectable changes within the white matter tracts evaluated with DTI [118].

#### 6.6.3. Distal Hereditary Motor Neuropathies

Distal hereditary motor neuropathies (dHMN) are very rare, genetically and clinically heterogeneous lower motor neuron diseases that can be inherited in an autosomal dominant, autosomal recessive or X-chromosome-coupled manner. So far, the mutation of more than a dozen genes responsible for the development of the disease has been discovered. Mutation of a single gene can cause different phenotypes of the disease. However, it is estimated that in 80% of cases dHMNs are caused by a mutation of a yet undiscovered gene. Most patients develop symptoms within the first two decades of life. Hereditary motor neuropathies are characterized by slowly progressive, symmetrical and mainly distal paresis of the lower limbs, without accompanying sensory disturbances. In dHMN, there are no bulbar symptoms. The development of symptoms occurs in childhood, with symptoms usually beginning in the feet (hollow foot, flat foot, hammer toes). The exception is type V according to Harding, in which the first symptoms develop in the upper limbs. In addition to limb paresis, some types of dHMN have associated symptoms, such as diaphragmatic paralysis or vocal cord paralysis [119,120,121]. An important role in the diagnosis of dHMN is served by electrophysiological examination, which allows for the differentiation of this group of diseases with genetically determined motor-sensory polyneuropathies (CMT—Charcot–Marie–Tooth disease) and distal myopathies. Nerve conduction studies show reduced amplitudes of motor responses; the changes are dependent on fiber length. In contrast to CMT, sensory conduction is normal. Electromyographic examination confirms chronic neurogenic damage to distal muscles [119,120,121].

#### 6.6.4. Hereditary Spastic Paraplegia

Hereditary spastic paraplegia (HSP) is a group of diseases in which damage to the upper motor neuron occurs. This is a heterogeneous group of disorders that can be inherited in an autosomal dominant, autosomal recessive, X-chromosome-coupled or mitochondrial manner or as a result of sporadic mutations. Also, considerable heterogeneity within families has been demonstrated. To date, more than 80 genes responsible for the development of HSP have been discovered. The prevalence is 1.8/100,000 [122,123]. The disease progresses slowly. Spastic paraparesis inherited in an autosomal dominant manner usually develops in adulthood (between the second and third decade of life), while those inherited in an autosomal recessive manner develop in childhood or young adulthood. The main symptom of HSP is spastic paresis of the lower limbs, which results from the damage to corticospinal tracts. It may be accompanied by sphincter or deep sensory disturbances [122,123,124,125,126,127].

HSP is divided into simple (paresis is the only symptom) and complex (concomitant involvement of other organs) forms. In the complex form of HSP, there may be involvement of the cerebellum, extrapyramidal system, peripheral nerves or muscles. Cognitive or mental dysfunction may also be present in this group of patients. Sometimes patients have dysmorphic features (facial changes, short stature), skeletal malformations (scoliosis, malformations in the feet) or involvement of the visual system (cataracts, retinal pigmentary degeneration, optic nerve atrophy). Imaging studies of patients with complex HSP show changes in the white matter of the brain, thinning of the corpus callosum and atrophy of the spinal cord or cerebellum [122,124,125].

Table 2 shows the summary of electrodiagnostic criteria for hereditary motor neuropathies.

## 7. Conclusions

The substantial progress in modern diagnostic techniques and pharmacotherapy, facilitated by a progressively more detailed understanding of the molecular basis of drug mechanisms, has significantly increased the effectiveness of treatment across many dis-eases. Consequently, establishing an accurate diagnosis at the earliest possible stage re-mains critical for optimal patient management. This manuscript summarizes the phenotypic variants of MND and outlines the currently accepted diagnostic criteria. It also re-views conditions that require careful differentiation from MND, with emphasis on the role of electrophysiological testing. Electromyography and related neurophysiological assessments are highlighted as essential tools for confirming motor neuron involvement and distinguishing MND from its clinical mimics.

## Figures and Tables

**Figure 1 jcm-15-00287-f001:**
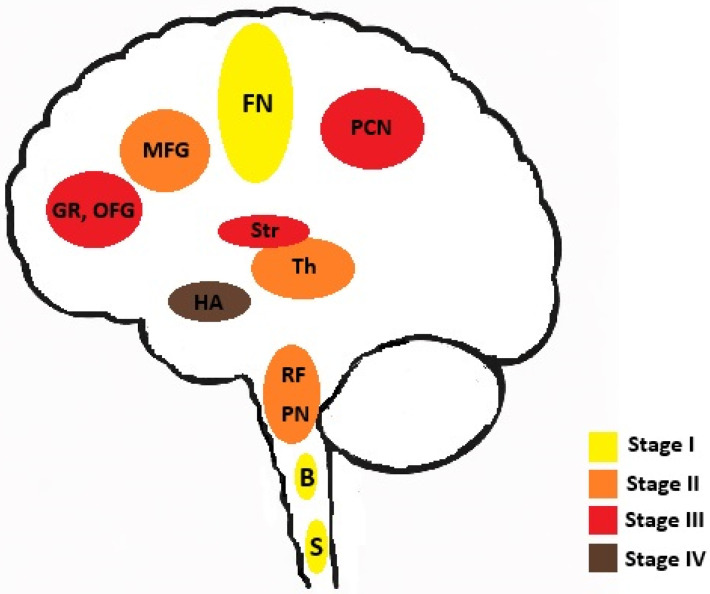
Sequential distribution of pTDP-43 pathology in amyotrophic lateral sclerosis according to Braak’s corticofugal axonal spread model. Each stage of the disease progression is represented by a distinctive color. Abbreviations: FN—frontal neocortex, B—bulbar somatomotor neurons, S—spinal motor neurons, RF—reticular formation, PN—precerebellar nuclei, Th—thalamus, MFG—middle frontal gyrus, GR—gyrus rectus, OFG—orbital frontal gyri, PCN—postcentral neocortex, Str—Striatum.

**Figure 2 jcm-15-00287-f002:**
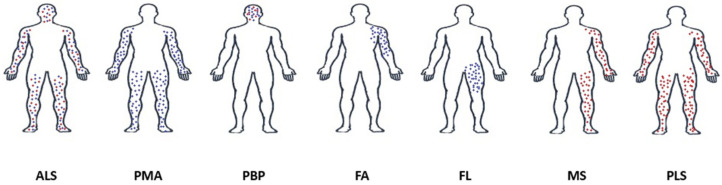
Schematic representation of motor neuron disease variants, illustrating the typical distribution of clinical and electrophysiological findings. Upper motor neuron (UMN) involvement is shown in red; lower motor neuron (LMN) involvement is shown in blue. Abbreviations: ALS—amyotrophic lateral sclerosis, PMA—progressive muscular atrophy, PBP—progressive bulbar palsy, FA—flail arm syndrome, FL—flail leg syndrome, MS—Mills syndrome, PLS—primary lateral sclerosis.

**Figure 3 jcm-15-00287-f003:**
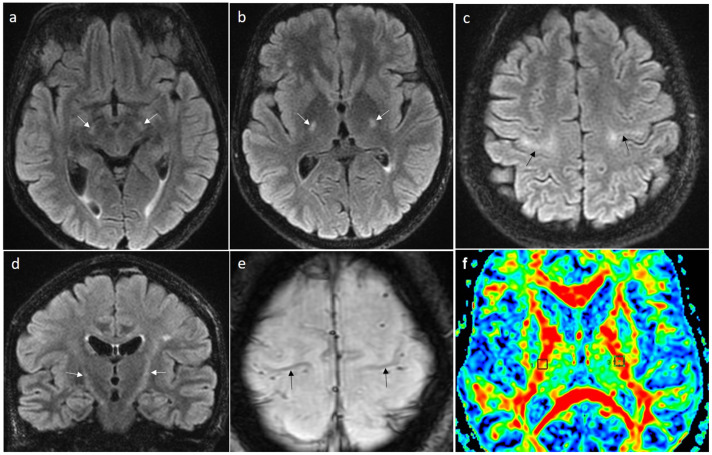
Magnetic resonance images of a patient with SLA. Axial FLAIR images showing hyperintense signal within both pyramidal tracts at the level of cerebral peduncles (**a**), posterior limbs of internal capsules (**b**) and precentral gyri (**c**), also seen in a coronal view (**d**) (arrows). “Motor band sign” visible on SWI as a hypointense line of iron deposition bilaterally along the primary motor cortex (**e**) (arrows). A DTI map (**f**) of Fractional Anisotropy (FA) with two square Regions of Interest within posterior limbs of both internal capsules showing low FA values of 0.513 on the right side and 0.580 on the left side (normal values should be above 0.650) indicating white matter disintegrity (own material).

**Figure 4 jcm-15-00287-f004:**
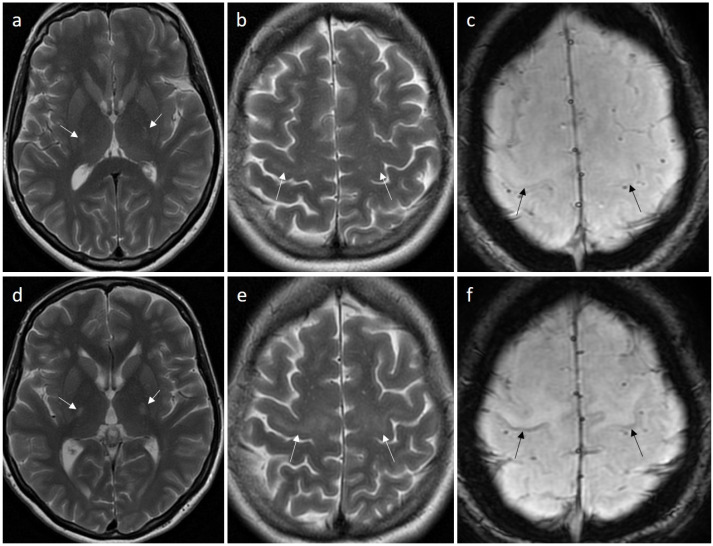
Magnetic resonance images of a patient with SLA. Initial study (upper row) and a follow-up study (lower row) 9 months later showing a pronounced progression of imaging symptoms of hyperintensity on T2-weighted images along the pyramidal tracts (**a**,**d**) and in the precentral gyri (**b**,**e**) as well as the worsening of the motor band sign in the primary motor cortex on SWI (**c**,**f**).

**Table 1 jcm-15-00287-t001:** Comparative overview of the revised El Escorial, Awaji and Gold Coast diagnostic criteria for ALS.

	El Escorial Criteria (Revised) [26]	Awaji Criteria [28]	Gold Coast Criteria [30]
Rationale for introducing criteria	Establishing classification for research and clinical purposes	Improving sensitivity and strengthening the role of electrophysiological findings	Improving sensitivity, simplifying the classification
Sensitivity	Lower	Higher	Highest
Specificity	Highest	High	Lower
Structure	Definite ALS: clinical presence of UMN and LMN signs in three regions Probable ALS: clinical UMN and LMN signs in at least two regions, some UMN signs above the LMN signs Probable ALS—Laboratory-supported: clinical UMN and LMN signs in one region or UMN signs alone in one region, and LMN signs defined by EMG criteria in at least two regions Possible ALS: clinical signs of UMN and LMN dysfunction in one region or UMN signs alone in two or more regions; or LMN signs rostral to UMN signs	Definite ALS: clinical or electrophysiological presence of UMN and LMN signs in three regions Probable ALS: clinical or electrophysiological UMN and LMN signs in at least two regions, some UMN signs above the LMN signs Possible ALS: clinical or electrophysiological signs of UMN and LMN dysfunction in one region or UMN signs alone in two or more regions; or LMN signs rostral to UMN signs	The presence of UMN and LMN dysfunction in at least one body region or LMN dysfunction in at least two body regions
Electrophysiological criteria	Signs of active denervation: 1. fibrillation potentials 2. positive sharp waves Signs of chronic denervation: 1. large motor unit potentials of increased duration with an increased proportion of polyphasic potentials, often of increased amplitude 2. reduced interference pattern with fringe rates higher than 10 Hz unless there is a significant UMN component 3. unstable motor unit potentials	Similar to El Escorial criteria + fasciculation potentials, preferably of complex morphology, equivalent to fibrillations and positive sharp waves	Similar to Awaji-shima criteria

**Table 2 jcm-15-00287-t002:** Characterization of electrodiagnostic criteria for hereditary motor neuropathies that mimic ALS. Abbreviations: spinal muscular atrophy (SMA), spinal and bulbar muscular atrophy (SBMA), distal hereditary motor neuropathies (dHMN), sigma non-opioid intracellular receptor 1 gene (SIGMAR1), hereditary spastic paraplegia (HSP), needle electromyography (EMG), MU potential (MUP).

Electrodiagnostic Criteria		Motor Nerve Conduction Criteria	Sensory Nerve Conduction Criteria	Needle Electromyography (EMG)	
				Spontaneous Activity	MUP
SMA	SMA 1, 2	Normal or slightly reduced amplitude of CMAP	Normal	Fibrillation potentials, positive sharp waves, fasciculation potentials	Large, rapidly firing with a late recruitment and reduced interference pattern Reinnervation more common in SMA 2 than SMA1
	SMA 3, 4	Reduced CMAP	Normal	Fasciculation potential occurs infrequently Complex repetitive discharges suggest a late stage	A late MUP recruitment reflects the loss of anterior horn cells High amplitude, long- duration MUP, reinnervation
SBMA		Reduced CMAP	Absent or low amplitude SNAP	Fibrillation potentials, CRD, reduced values of MUNE	Large MUP, long duration with reduced recruitment
dHMN	SIGMAR1-related dHMN with pyramidal signs	Significantly reduced CMAP	Normal	Denervated potentials localized symmetrically	Large MUP
	Charcot–Marie–Tooth type 4 Autosomal Recessive	Mostly demyelinating NCV < 38 m/s			
	Charcot–Marie–Tooth type 2, Autosomal Dominant	Reduction in amplitude mild slowing NCV > 38 m/s	Mild reduction in amplitude	Fasciculation potentials, fibrillation potentials, positive sharp waves	Large long duration MUAPs with early recruitment
	Charcot–Marie–Tooth type 1 Autosomal Dominant	Demyelination NCV < 38 m/s temporal dispersion is typically not seen		Fasciculation potentials, fibrillation potentials, positive sharp waves A waves	Large, long duration MUAPs with early recruitment
	Charcot–Marie–Tooth X-linked	Demyelinative features but mostly distal motor nerve fibers with primary axonal degeneration mostly NCV >38	Demyelinative features but mostly distal sensory nerve fibers/axonal degeneration	Fasciculation potentials, fibrillation potentials, positive sharp waves	Large, long duration MUAPs with early recruitment
HSP—Hereditary spastic paraplegia		Mostly normal, depends on type HSP axonal/demyelinating	Mostly normal	Normal, fasciculation potentials might be seen	Large, long duration MUAPs with early recruitment

## Data Availability

Data sharing not applicable to this article as no datasets were generated or analyzed during the current study.

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
