# Peer review of "Establishing Diagnostic and Differential Diagnostic Criteria for Amyotrophic Lateral Sclerosis"

_jcm, 2025, doi:10.3390/jcm15010287_

Round 1
Reviewer 1 Report
Comments and Suggestions for Authors
- I would like to suggest authors the inclusion of a Table summarizing and comparing basic aspects between Gold Coast criteria, Awaji-shima criteria and El Escorial criteria.
- In the discussion of 5q SMA as a differential diagnosis of ALS, authors should include an additional and specific paragraph with specific literature related to SMA type 4, as they did for SMA type 3, as both types have different motorprogression and severity.
- In the Abstract, the description of FOSMN and FEWDON syndromes is not correctly presented by the authors and should be reviewed.
- There is a minor typo in line 372 (“haemiplegic”) that should be corrected.
- As the authors considered the differential diagnosis of monomelic amyotrophy, I suggest authors to consider also the inclusion of O’Sullivan-McLeod syndrome as an important differential diagnosis in cases with bilateral involvement and mainly upper limbs involvement. They could consider the inclusion of a topic after the “Monomelic Amyotrophy” (6.2) with specific references, for example.
- The authors have included both SBMA and dHMN in the Discussion of differential diagnosis, however they have not included a specific topic for non-5q SMA, which may represent in several cases one of the most important differential diagnosis. I suggest the inclusion of this topic before the Hereditary Spastic Paraplegia topic in the text and the inclusion of references related to this topic.
- I suggest authors to consider the inclusion of a conclusion topic at the final of their manuscript, because in the current format of the text the content ends after Table 1.
Author Response
We highly appreciate the Reviewer’s keen and thorough remarks, aimed at improvement of the comprehensive quality of the manuscript, as well as clear organization of its content. In the revised version of the manuscript, we tried our best to consider all the issues raised in the review.
Due to the inclusion of additional paragraphs, the numbering of references has changed.
Particular concerns of the Reviewer are point-by-point addressed below:
Reviewer 1:
- I would like to suggest authors the inclusion of a Table summarizing and comparing basic aspects between Gold Coast criteria, Awaji-shima criteria and El Escorial criteria.
Thank you for this helpful suggestion. We agree that a comparative overview of the diagnostic criteria could improve clarity for readers. In the revised manuscript, we have added a new table (Table 1) that summarizes the key features of the Gold Coast, Awaji-Shima, and revised El Escorial criteria.
|
|
El Escorial criteria (revised)26 |
Awaji-Shima criteria28 |
Gold Coast criteria30 |
|
Rationale for introducing criteria |
Establishing classification for research and clinical purposes |
Improving sensitivity and strenghtening the role of electrophysiological findings |
Improving sensitivity, simplifying the classification |
|
Sensitivity |
Lower |
Higher |
Highest |
|
Specificity |
Highest |
High |
Lower |
|
Structure |
Definite ALS: clinical presence of UMN and LMN signs in three regions
Probable ALS: clinical UMN and LMN signs in at least two regions, some UMN signs above the LMN signs
Probable ALS – Laboratory-supported: clinical UMN and LMN signs in one region or UMN signs alone in one region, and LMN signs defined by EMG criteria in at least two regions
Possible ALS: clinical signs of UMN and LMN dysfunction in one region or UMN signs alone in two or more regions; or LMN signs rostral to UMN signs |
Definite ALS: clinical or electrophysiological presence of UMN and LMN signs in three regions
Probable ALS: clinical or electrophysiological UMN and LMN signs in at least two regions, some UMN signs above the LMN signs
Possible ALS: clinical or electrophysiological signs of UMN and LMN dysfunction in one region or UMN signs alone in two or more regions; or LMN signs rostral to UMN signs |
The presence of UMN and LMN dysfunction in at least one body region or LMN dysfunction in at least two body regions |
|
Electrophysiological criteria |
Signs of active denervation: 1. fibrillation potentials 2. positive sharp waves
Signs of chronic denervation: 1. large motor unit potentials of increased duration with an increased proportion of polyphasic potentials, often of increased amplitude 2. reduced interference pattern with fring rates higher than 10 Hz unless there is a signifcant UMN component 3. unstable motor unit potentials |
Similar to El Escorial criteria + fasciculation potentials, preferably of complex morphology, equivalent to fibrillations and positive sharp waves |
Similar to Awaji-Shima criteria |
Table 1. Comparative overview of the revised El Escorial, Awaji-Shima and Gold Coast diagnostic criteria for ALS.
- In the discussion of 5q SMA as a differential diagnosis of ALS, authors should include an additional and specific paragraph with specific literature related to SMA type 4, as they did for SMA type 3, as both types have different motorprogression and severity.
Thank you for this helpful suggestion. We have reorganized and expanded the relevant section of the Discussion to more clearly emphasize the differences between different SMA types.
- In the Abstract, the description of FOSMN and FEWDON syndromes is not correctly presented by the authors and should be reviewed.
Thank you for pointing this out. We reviewed the Abstract and revised the wording to ensure accurate decription.
- There is a minor typo in line 372 (“haemiplegic”) that should be corrected.
The word “haemiplegic” has been corrected to “hemiplegic” in the revised manuscript.
- As the authors considered the differential diagnosis of monomelic amyotrophy, I suggest authors to consider also the inclusion of O’Sullivan-McLeod syndrome as an important differential diagnosis in cases with bilateral involvement and mainly upper limbs involvement. They could consider the inclusion of a topic after the “Monomelic Amyotrophy” (6.2) with specific references, for example.
Thank you for this valuable suggestion. We agree that O’Sullivan–McLeod syndrome represents an important differential diagnosis, particularly in cases with bilateral involvement and predominant upper-limb weakness. In response, we have added a dedicated paragraph following the section on Monomelic Amyotrophy, as suggested by the Reviewer.
- The authors have included both SBMA and dHMN in the Discussion of differentia diagnosis, however they have not included a specific topic for non-5q SMA, which may represent in several cases one of the most important differential diagnosis. I suggest the inclusion of this topic before the Hereditary Spastic Paraplegia topic in the text and the inclusion of references related to this topic.
Thank you for this important observation. We have added a dedicated paragraph on non-5q SMA in the SMA section.
- I suggest authors to consider the inclusion of a conclusion topic at the final of their manuscript, because in the current format of the text the content ends after Table 1.
Thank you for this helpful comment. We agree that adding a concluding section improves the structure and readability of the manuscript. Therefore, we have added a Conclusions section at the end of the text.
Reviewer 2 Report
Comments and Suggestions for Authors
The manuscript by Edyta Dziadkowiak et al. is a very interesting narrative review that provides an up-to-date, easily accessible summary of the MND spectrum and useful guidelines for differential diagnosis. I believe it can be a useful contribution to the literature, particularly for educational purposes. I have only a few suggestions for the Authors:
- It is not clear how this work differs from many others available in the literature. I suggest that, at the end of the introduction, you explain the objectives and how they differ from other works in the literature, perhaps citing recent reviews to highlight these differences.
- In the Methods section, if this is a narrative review, the level of methodological detail provided is excessive. I suggest summarising or, if you prefer to keep it, being more transparent about the criteria chosen for including or excluding works (what is meant by the “relevance” of the paper?);
- Many neurophysiological findings are not clearly defined or explained, such as signs of denervation. I suggest providing a brief definition of each neurophysiological biomarker to improve the reader's understanding;
- I recommend exploring the role that AI could play in helping to diagnose this complex condition, for example, by citing and discussing works such as doi:10.1080/21678421.2024.2334836.
- Although Hirayama's disease is prevalent in Eastern populations, it can also be found in Western countries. I suggest citing and discussing this work doi:10.3389/fneur.2020.00183 to emphasise the possibility of identifying it even in non-“typical” patients.
- With regard to MMN, I suggest citing the efficacy of certain drugs, e.g., cyclophosphamide, to highlight their usefulness for differential diagnosis; 10.1007/s11940-013-0269-y
Author Response
We highly appreciate the Reviewer’s keen and thorough remarks, aimed at improvement of the comprehensive quality of the manuscript, as well as clear organization of its content. In the revised version of the manuscript, we tried our best to consider all the issues raised in the review.
Due to the inclusion of additional paragraphs, the numbering of references has changed.
Particular concerns of the Reviewer are point-by-point addressed below:
- It is not clear how this work differs from many others available in the literature. I suggest that, at the end of the introduction, you explain the objectives and how they differ from other works in the literature, perhaps citing recent reviews to highlight these differences.
Thank you for this insightful comment. We have revised the end of the Introduction to clearly state the objectives of our review and to highlight how our approach differs from previously published works. Specifically, we now emphasize that our manuscript provides a practical discussion of electrophysiological criteria for differentiating MND, together with an expanded and clinically oriented overview of the differential diagnosis of ALS.
- In the Methods section, if this is a narrative review, the level of methodological detail provided is excessive. I suggest summarising or, if you prefer to keep it, being more transparent about the criteria chosen for including or excluding works (what is meant by the “relevance” of the paper?)
Thank you for this suggestion. In accordance with your comment, we have shortened and summarised the Methods section to reflect the narrative nature of the review.
„A literature search, covering the period of 1990-2025, was performed using the Pub-Med and Embase databases. Due to the narrative character of the review, no strict, formal search strategy or study selection protocol were applied. The selection intended to include representative publications that inform current clinical practice.”
- Many neurophysiological findings are not clearly defined or explained, such as signs of denervation. I suggest providing a brief definition of each neurophysiological biomarker to improve the reader’s understanding
Thank you for this constructive comment. In response, we have expanded Section 5.1 to include brief explanatory descriptions of the neurophysiological terms and biomarkers discussed, including signs of denervation. These additions aim to improve clarity and understanding for readers who may not have a specialized background in clinical neurophysiology.
- I recommend exploring the role that AI could play in helping to diagnose this complex condition, for example, by citing and discussing works such as doi:10.1080/21678421.2024.2334836.
Thank you for this helpful suggestion. We have now added a concise paragraph discussing the emerging role of artificial intelligence in ALS diagnosis at the end of 5.1 section.
- Although Hirayama’s disease is prevalent in Eastern populations, it can also be found in Western countries. I suggest citing and discussing this work doi:10.3389/fneur.2020.00183 to emphasise the possibility of identifying it even in non-“typical” patients.
Thank you for this valuable suggestion. We have included and discussed the recommended case report (doi:10.3389/fneur.2020.00183) in the relevant section.
- With regard to MMN, I suggest citing the efficacy of certain drugs, e.g., cyclophosphamide, to highlight their usefulness for differential diagnosis; 10.1007/s11940-013-0269-y
Thank you for this helpful comment. We have now incorporated a short paragraph regarding the reported efficacy of cyclophosphamide and other immunomodulatory/immunosupressive agents in MMN and added the recommended reference.
Round 2
Reviewer 1 Report
Comments and Suggestions for Authors
- Reference 92 is the same as reference 6.
- Reference 105 added by the authors does not have any direct correlation with non-5q SMA.
- There are minor typos that should be corrected in the text (e.g. line 103, “di-agnostic”).
Author Response
We would like to thank the reviewers and the editor once again for their valuable feedback and for the opportunity to further revise our manuscript. We appreciate that the majority of the concerns have been resolved during the first revision. In this second round, we have addressed the three remaining comments below:
- Reference 92 is the same as reference 6.
Thank you for pointing this out. Reference 92 was a duplicate of reference 6 and has now been corrected.
- Reference 105 added by the authors does not have any direct correlation with non-5q SMA.
We appreciate the reviewer’s observation. Reference 105 (now 104) has been replaced with a more appropriate references addressing this topic of non-5q-SMA (references 112,113).
- There are minor typos that should be corrected in the text (e.g. line 103, “di-agnostic”).
Minor typographical errors throughout the manuscript, including the one noted in line 103 have been corrected.

Reviewer 2 Report
Comments and Suggestions for Authors
I thank the Authors for their extensive work, which significantly improved the quality of their manuscript. No further comments
